# Predicting the Impact of Polysulfone Dialyzers and Binder Dialysate Flow Rate on Bilirubin Removal

**DOI:** 10.3390/bioengineering11121262

**Published:** 2024-12-12

**Authors:** Alexander Novokhodko, Nanye Du, Shaohang Hao, Ziyuan Wang, Zhiquan Shu, Suhail Ahmad, Dayong Gao

**Affiliations:** 1Mechanical Engineering, University of Washington (Seattle), 3900 E Stevens Way NE, Seattle, WA 98195-0001, USA; novalex@uw.edu (A.N.); dunanye1@uw.edu (N.D.); shaohang@uw.edu (S.H.); wangzi@uw.edu (Z.W.); 2School of Engineering and Technology, University of Washington (Tacoma), 1900 Commerce Street, Tacoma, WA 98402-3100, USA; zqshu@uw.edu; 3School of Medicine, University of Washington (Seattle), 1959 N.E. Pacific Street-Box 356340, Seattle, WA 98195-6340, USA

**Keywords:** liver, albumin, dialysis, liver dialysis, albumin dialysis, computational modeling, ordinary differential equations, liver failure

## Abstract

Liver failure is the 12th leading cause of death worldwide. Protein-bound toxins such as bilirubin are responsible for many complications of the disease. Binder dialysis systems use albumin or another binding molecule in dialysate and detoxifying sorbent columns to remove these toxins. Systems like the molecular adsorbent recirculating system and BioLogic-DT have existed since the 1990s, but survival benefits in randomized controlled trials have not been consistent. New binder dialysis systems, including open albumin dialysis and the Advanced Multi-Organ Replacement system, are being developed. Optimal conditions for binder dialysis have not been established. We developed and validated a computational model of bound solute dialysis. It predicted the impact of changing between two test setups using different polysulfone dialyzers (F3 and F6HPS). We then predicted the impact of varying the dialysate flow rate on toxin removal. We found that bilirubin removal declines with dialysate flow rate. This can be explained through a linear decline in free bilirubin membrane permeability. Our model quantifies this decline through a single parameter (polysulfone dialyzers). Validation for additional dialyzers and flow rates will be needed. This model will benefit clinical trials by predicting optimal dialyzer and flow rate conditions. Accounting for toxin adsorption onto the dialyzer membrane may improve results further.

## 1. Introduction

In 2021, liver failure was the 12th leading cause of death globally [1]. The liver performs a variety of critical functions, including toxin removal, drug and food metabolism, protein synthesis, including synthesis of clotting factors and immunoproteins, nutrient storage, and other functions. The buildup of toxins during liver failure drives hepatic coma, liver cell death, and failure of other organ systems. Bilirubin is a commonly studied hepatic toxin. Bilirubin accumulation is visible in the skin as jaundice. Excess bilirubin contributes to the pathology of liver failure, for example, by damaging white matter [2], which may contribute to hepatic coma. It is also a commonly accepted marker for other albumin-bound toxins.

Bilirubin cannot be removed by traditional dialysis because it is a hydrophobic molecule that is bound to albumin in human blood [3]. Adding a binder such as albumin or charcoal to the dialysate allows bilirubin removal [4]. Computational modeling of albumin dialysis was first carried out in the seminal work of Patzer and colleagues, who described single-pass albumin dialysis (SPAD) in three publications [4,5,6]. Today, the only FDA-approved albumin dialysis system is the Molecular Adsorbent Recirculating System (MARS) [7], which recirculates albumin instead of discarding it after a single pass. Three models of closed-loop mode albumin dialysis have been published and are summarized in Table 1.

The first is by Magosso and colleagues [8]. This model was designed using clinical data to model MARS sessions. It accounts for ultrafiltration and diffusion. However, it assumes that the dialyzer blood and dialysate compartments and charcoal and resin columns of MARS are well-mixed compartments without modeling concentration gradients. The model has not been validated against data beyond what was used to fit its parameters.

The second is by Annesini and colleagues [9,10,11,12,13]. This model applies chemical engineering techniques. It cannot be applied to bilirubin because it assumes albumin concentration to be far greater than toxin concentration. During hyperbilirubinemia in liver failure, the concentration of bilirubin may be as high as 809.8 µM, while the concentration of albumin may be as low as 144 µM [18]. In this case, that assumption is violated. Schiesser [19] explains the implementation of this model.

The third is by Pei and colleagues, who extended the work of Patzer to include recirculation and the effect of local ultrafiltration [14,15,16,17]. In this work, their model could not be replicated (see Methods and Results below). Since their code is not publicly available, their equation for ultrafiltration rate in the Peclet number formula is not reported, making it difficult to identify why their data could not be reproduced. The replication attempt presented in this manuscript used Villarroel’s definition of the Peclet Number [20].

MARS has not shown consistent survival improvements in randomized controlled trials [21,22]. Today, a new generation of binder dialysis systems is being developed. These include the Advanced Multi-Organ Replacement system (AMOR) [23], Open Albumin Dialysis (OPAL) [24], and High-Efficiency MARS (HE-MARS) [25]. Our goal is to optimize AMOR using this algorithm, but any binder dialysis system could be modeled in this way.

## 2. Materials and Methods

### 2.1. Attempt to Replicate the Model of Pei and Colleagues

The first step was attempted to replicate the model given by Pei and Colleagues using the parameter values and equations they provide for the Gambro 6LR dialyzer [14,15,16,17]. Since their manuscripts do not define the Peclet number, Villarroel’s definition of the Peclet Number was used [20]. This replication work was undertaken using Matlab 2020. The Regula Falsi method was used to solve the boundary value problem (BVP). Because their work could not be reproduced (see Section 3), a modified model was created.

### 2.2. Model Description

A new model was constructed similarly to that of Pei and colleagues [14,15,16,17]. Blood and dialysate were assumed to be two well-mixed compartments containing albumin and the toxin of interest. Toxin removal was assumed to happen by diffusion and convection across a dialyzer membrane with blood and dialysate flowing counter-current. The removal of toxin across the dialyzer was solved at each time step by solving a spatial model of concentrations, pressures, and flow rates. The spatial model was one-dimensional, with the distance along the axis from the blood inlet to the blood outlet defined as z. This was used to update the blood and dialysate concentrations. The Peclet number was defined following previous work [20]. Unlike the model of Pei and colleagues, a piecewise function was used to account for the change in the sign of convective flux across the dialysis membrane during backfiltration.

### 2.3. Variable Names

$P_{b}\to$ blood pressure, $P_{d}\to$ dialysate pressure, *z* → distance along dialyzer fiber from blood inlet and dialysate outlet. $\mu_{b}\to$ viscosity of blood. $\mu_{d}\to$ viscosity of dialysate. $Q_{b}\to$ blood flow rate. $Q_{d}\to$ dialysate flow rate. $r_{i}\to$ inner dialysate fiber radius. $r_{o}\to$ outer dialysate fiber radius. $R_{m}\to$ inner radius of dialyzer shell (constrains the dialysate space outside the fibers). n→ number of fibers. $L_{p}\to$ Membrane hydraulic permeability. $C_{stlb}\to$ total blood bilirubin concentration. $C_{stld}\to$ total dialysate bilirubin concentration. $C_{sb}\to$ free blood bilirubin concentration. $C_{sd}\to$ free dialysate bilirubin concentration. $C_{atlb}\to$ total blood albumin. $C_{atld}\to$ total dialysate albumin. $K_{free}A$ → a coefficient representing the diffusive transport of unbound bilirubin across the entire area of the dialyzer membrane. $K_{B}\to$ albumin–bilirubin binding constant, $J_{v}\to$ local ultrafiltration flux. *f* → function of the Peclet number. σ→ reflection coefficient of the membrane. *L* → dialyzer length. *pe* → the Peclet number, a dimensionless quantity relating convection and diffusion. $V_{b}\to$ blood reservoir volume. $V_{d}\to$ dialysate reservoir volume. χ → amount of toxin removed from blood to dialysate, with a negative sign. $x_{y0\;}\to$ negative amount removed over the entire single pass through the dialyzer. $K_{free}A_{Qd=500}\to$ The theoretical value of $K_{free}A$ at a dialysate flow rate of 500 mL/min. $\beta_{Q_{d}}\to$ The percent change in $K_{free}A$ from 500 to 800 mL/min. UCF → a unit conversion factor constant.

### 2.4. Boundary Conditions

$P_{d}\left(z=0\right)=0$ (dialysate outlet pressure), $C_{stlb}\left(z=0,t=0\right)=C_{stlb,in,0}$ (input blood side bilirubin). $C_{stld}\left(z=L,\;t=0\right)=0$ (dialysate initially has no bilirubin). $Q_{b}\left(z=L\right)=Q_{b}\left(z=0\right)$ (no net ultrafiltration). $Q_{d}\left(z=L\right)=Q_{d}\left(z=0\right)$ (no net ultrafiltration). $C_{atlb}\left(z=0\right)=C_{atlb,in}$ (constant blood inlet albumin concentration). $C_{atlb}\left(z=L\right)=C_{atlb,\;out}$ (constant blood outlet albumin concentration). $Q_{b}\left(z=0\right)=Q_{b,in}$ (Constant inlet blood flow rate). $Q_{b}\left(z=L\right)=Q_{b,out}$ (constant outlet blood flow rate). $P_{b}\left(z=0\right)=P_{b,in}$ (constant inlet blood pressure). $P_{b}\left(z=L\right)=P_{b,out}$ (constant outlet blood pressure). $C_{stlb}\left(z=0,\;t\right)=C_{stlb,in}\left(t\right)$ (time-dependent blood inlet toxin concentration boundary condition). $C_{stlb}\left(z=L,\;t\right)=C_{stlb,out}\left(t\right)$ (time-dependent blood outlet toxin concentration boundary condition). $C_{atld}\left(z=L\right)=C_{atld,in}$ (constant dialysate inlet albumin concentration). $C_{atld}\left(z=0\right)=C_{atld,\;out}$ (constant dialysate outlet albumin concentration). $Q_{d}\left(z=L\right)=Q_{d,in}$ (constant inlet dialysate flow rate). $Q_{d}\left(z=0\right)=Q_{d,out}$ (constant outlet dialysate flow rate). $P_{d}\left(z=L\right)=P_{d,in}$ (constant inlet dialysate pressure). $P_{d}\left(z=0\right)=P_{d,out}$ (constant outlet dialysate pressure). $C_{stld}\left(z=L,\;t\right)=C_{stld,in}\left(t\right)$ (time-dependent dialysate inlet toxin concentration boundary condition). $C_{stld}\left(z=0,\;t\right)=C_{stld,out}\left(t\right)$ (time-dependent dialysate outlet toxin concentration boundary condition). χ(z = 0) = 0 (no bilirubin removed at dialysate inlet). χ(z = L) = $x_{y0\;}$ (boundary condition for χ at z = L (dialysate inlet, blood outlet), used for solving the BVP).

Figure 1 shows the boundary conditions and variables of the model along with the direction of flow vectors. The condition of local transmembrane pressure dependent ultrafiltration and backfiltration with net zero ultrafiltration is illustrated in panel B.

### 2.5. Equations


(1)
$$\mathrm{Change}\;\mathrm{in}\;\mathrm{blood}\;\mathrm{pressure}\;\mathrm{with}\;\mathrm{respect}\;\mathrm{to}\;z:\frac{dP_{b}}{dz}=-\frac{8\mu_{b}Q_{b}}{n\pi r_{i}^{4}}$$


Equation (1) from Equation (2) in [17].
(2)$$\mathrm{Change}\;\mathrm{in}\;\mathrm{dialysate}\;\mathrm{pressure}\;\mathrm{with}\;\mathrm{respect}\;\mathrm{to}\;z:\;\frac{dP_{d}}{dz}=\frac{8\mu_{d}{\left(R_{m}+nr_{o}\right)}^{2}Q_{d}}{\pi {\left(R_{m}^{2}-nr_{0}^{2}\right)}^{3}}$$

Equation (2) from Equation (3) in [17].
(3)$$\mathrm{Change}\;\mathrm{in}\;\mathrm{blood}\;\mathrm{flow}\;\mathrm{rate}\;\mathrm{with}\;\mathrm{respect}\;\mathrm{to}\;z:\frac{dQ_{b}}{dz}=-2n\pi r_{i}L_{p}\left(P_{b}-P_{d}\right)$$

Equation (3) from Equations (4) and (6) in [17].
(4)$$\mathrm{Change}\;\mathrm{in}\;\mathrm{dialysate}\;\mathrm{flow}\;\mathrm{rate}\;\mathrm{with}\;\mathrm{respect}\;\mathrm{to}\;z:\frac{dQ_{d}}{dz}=-2n\pi r_{i}L_{p}\left(P_{b}-P_{d}\right)$$

Equation (4) from Equations (5) and (6) in [17].

Change in blood and dialysate total bilirubin concentration multiplied by flow rate with respect to *z*: this equation is modified from Equations (7) and (8) [17]. Unlike the previous model, the change in the direction of convection across the dialyzer membrane when dialysate pressure exceeds blood pressure is accounted for. A purely diffusive mode is introduced for small values of $J_{v}$ to avoid division by zero errors when calculating *f* (see below).
(5)$$\frac{d\left(Q_{b}C_{stlb}\right)}{dz}=\frac{d\left(Q_{d}C_{stld}\right)}{dz}=\;\left\{\begin{array}{c} \left(J_{v}\left(1-\sigma \right)f-\frac{K_{free}A}{L}\right)\times \left(C_{sb}-C_{sd}\right)-J_{v}\left(1-\sigma \right)C_{sb}\;if\;J_{v}>J_{v,crit}\\ -\frac{K_{free}A}{L}\times \left(C_{sb}-C_{sd}\right)\;if-J_{v,crit}<J_{v}<J_{v,crit}\\ \left(J_{v}\left(1-\sigma \right)f-\frac{K_{free}A}{L}\right)\times \left(C_{sb}-C_{sd}\right)-J_{v}\left(1-\sigma \right)C_{sd}\;if\;J_{v}<-J_{v,crit} \end{array}\right.$$

$J_{v,crit}$ is calculated by the following criterion:(6)$$J_{v,crit}=\min\left(Q_{b},Q_{d}\right)\times 10^{-5}\frac{1}{\mathrm{mm}}$$

Blood-free bilirubin in terms of blood total bilirubin:(7)$$C_{sb}=\frac{-\left(C_{atlb}+\frac{1}{K_{B}}-C_{stlb}\right)+\sqrt{{\left(C_{atlb}+\frac{1}{K_{B}}-C_{stlb}\right)}^{2}+\frac{4C_{stlb}}{K_{B}}}}{2}$$

Equation (7) from Equation (9a) [17].

Dialysate-free bilirubin in terms of dialysate total bilirubin:(8)$$C_{sd}=\frac{-\left(C_{atld}+\frac{1}{K_{B}}-C_{stld}\right)+\sqrt{{\left(C_{atld}+\frac{1}{K_{B}}-C_{stld}\right)}^{2}+\frac{4C_{stld}}{K_{B}}}}{2}$$

Equation (8) from Equation (9b) in [17].
(9)$$J_{v}=2n\pi r_{i}\;L_{p}\left(P_{b}-P_{d}\right)$$

Equation (9) from local ultrafiltration flux, Equation (6) in [17].
(10)$$f=\frac{1}{Pe}-\frac{1}{\exp\left(Pe\right)-1}$$

Equation (10) from the function of the Peclet number in [17].

The equation for the Peclet number is derived from Equation (9) in [20].
(11)$$Pe=J_{v}\times \left(1-\sigma \right)\times \frac{L}{kA}$$

Equation (11) from (9) in [20].

As the flow rate changes, albumin concentration also changes, but since the flow rate is periodic, so is the albumin concentration. Thus, it does not change over time, only spatially.
(12)$$C_{atlb}\times Q_{b}\left(z\right)=C_{atlb}\left(z=0\right)\times Q_{b}\left(z=0\right)$$

Equation (12) from (9b) in [17].
(13)$$C_{atld}\times Q_{d}\left(z\right)=C_{atld}\left(z=L\right)\times Q_{d}\left(z=L\right)$$

Equation (13) from (10b) in [17].

Time-dependence is described as follows:(14)$$\frac{d\left(C_{stlb,\;z=0}\times V_{br}\right)}{dt}=C_{stlb,out}\left(t\right)\times Q_{b,out}-C_{stlb,in}\left(t\right)\times Q_{b,in}$$

Equation (14) from (11a) in [17].
(15)$$\frac{d\left(C_{stld,\;z=0}\times V_{dr}\right)}{dt}=C_{stld,out}\left(t\right)\times Q_{d,out}-C_{stld,in}\left(t\right)\times Q_{d,in}$$

Equation (15) from (12a) in [17].

The impact of flow rate on dialyzer membrane permeability is modeled following previous work on renal dialysis [26].
(16)$$K_{free}A=K_{free}A_{Qd=500}\left(1+\beta_{Q_{d}}\times \frac{Q_{d,0}-500*UCF}{300*UCF}\right)$$

Equation (16) from (4) in [26]

Here, $K_{free}A_{Qd=500}$ is the theoretical mass transfer-area coefficient of unbound bilirubin diffusive transport at 500 mL/min. $\beta_{Q_{d}}$ is the percentage change in this coefficient, from 800 to 500 mL/min. It was set to 0.05544 during parameter fitting following past work on urea [26]. This indicates a 5.544% change in mass transfer area coefficient during a 300 mL/min change in dialysate side flow rate. UCF is a unit conversion factor. As an approximation, the adjustment is performed using the dialysate inlet and outlet flow rate $Q_{d,0}$ instead of the local dialysate flow rate $Q_{d}$. Modifying this equation and the governing equations of the model to account for the impact of local ultrafiltration on local $K_{free}A$ is planned in future work.

### 2.6. Fit Parameters

Three parameters were fit computationally rather than being determined from the literature. The first two were fit based only on F6HPS data at a single flow rate and used to predict the results for other conditions; these were $K_{B}$ and $K_{free}A_{Qd=500}$. The last parameter was initially set based on prior work and then updated once data for the F3 dialyzer at all dialysate flow rates was collected; this was $\beta_{Q_{d}}$.

The first was the primary binding constant for bilirubin–BSA binding ($K_{B}$). This parameter depends on conditions such as salt concentration, temperature, and pH [27,28]. No existing measurement of this parameter was conducted in dialysate at 37 °C, so the best available option is to approximate based on reported values under similar conditions and fit them to a test condition. All literature values for this parameter fall within a narrow range (Table 2). The primary binding site of bilirubin on BSA has approximately an order of magnitude greater affinity than the secondary binding site [29]. Thus, the secondary binding site was neglected for this analysis. The constant for the primary binding site was fit. The lower bound of the search space was set to 0.5 × 10^7^ (1/M). The upper bound was set to 7.5*10^7^ (1/M). The space was searched in steps of 1*10^7^ (1/M).

The second fit parameter was the product of the dialyzer mass transfer-area coefficient for free bilirubin diffusion and area at a dialysate flow rate of 500 mL/min ($K_{free}A_{Qd=500}$). This coefficient is analogous to *KoA* for toxins, which are not protein-bound, but there are important differences. This coefficient cannot be derived directly from the clearance of the toxin of interest by the standard relationship [35]. This calculation assumes that the mean concentration difference in counter-current dialysis is the logarithmic mean of the inlet and outlet concentration differences. For free bilirubin, this is not the case. On the blood side, as free bilirubin crosses the membrane, the unbound concentration declines. This creates a thermodynamic driving force for the release of additional bilirubin from albumin. In contrast, on the dialysate side, as bilirubin crosses the membrane, it binds to free binding sites on dialysate albumin. This lowers the free dialysate side bilirubin concentration. Additionally, this coefficient depends on the assumption that blood and dialysate side clearance are approximately equal. Other processes, such as membrane binding of bilirubin, may cause blood side clearance to exceed dialysate side clearance. Polysulfone membranes have been observed to bind small quantities of bilirubin [36]. This would lead to an apparently elevated $K_{free}A$ being the best fit parameter because it must account for other processes such as membrane binding.

This parameter varies between different dialyzers, different toxins, and different flow conditions [26]. Thus, it must be fit to experimental data for the specific toxin–dialyzer combination. The highest *KoA* value reported in the literature is 2000 mL/min [37]. Thus, this parameter was optimized over the range from 100 to 2500 mL/min in steps of 100 mL/min. This range is chosen because it encompasses the range typically reported for dialyzer membranes [38].

Afterwards, $K_{free}A_{Qd=500}$ was adjusted to account for a new dialyzer following Equation (20). Because the F3 and F6HPS dialyzers are both polysulfone, it was assumed that area was the only adjustment needed, meaning the two dialyzers had the same value of $K_{free}$.
(17)$$K_{free}A_{New,Qd=500}=K_{free}A_{Old,\;Qd=500}\times \frac{A_{new}}{A_{old}}.$$

The final fit parameter was the difference in $K_{free}A$ between 500 and 800 mL/min ($\beta_{Q_{d}}$). This parameter was initially set to 0.05544 based on past work on urea [26]. This variation is proposed to occur because of boundary layer effects. Once the results for different flow conditions were found, a parameter sweep of this value was conducted. The lower bound of this parameter sweep was 0, which represents the assumption that $K_{free}A$ is independent of dialysate side flow rate. The upper value was calculated as 0.6, since this is the value at which $K_{free}A\left(Q_{d}=0\right)=0$ mL/min. Values greater than this are not physically meaningful since $K_{free}A$ cannot be negative.

### 2.7. Modified Shooting Method to Ensure Numerical Stability

Because of the numerical instability, a shooting method was used to solve the boundary value problem posed by counter-current flow in the dialyzer. Details of the algorithm used to achieve numerical stability in solving this system are available in a provisional patent application [39] and it is discussed briefly in this section.

In addition to the modified shooting method, the ODEs that describe the system were rewritten in terms of the amount of toxin that moves from the blood to the dialysate side. This was defined using the variable χ.
(18)$$\chi =Q_{b}C_{stlb}-Q_{b,in}C_{stlb,in}=C_{stld}Q_{d}-C_{stld,out}Q_{d,in}.$$

Equation (18), definition of amount removed.

This variable has boundary conditions χ = 0 and z = 0 χ = $x_{y0}$ at z = L. The shooting method iterates over values of $x_{y0}$ defined as follows:(19)$$x_{y0}=C_{stlb,z=L}Q_{b,0}-C_{stlb,0}Q_{b,0}=C_{stld,z=L}Q_{d,0}-C_{stld,0}Q_{d,0}=\chi \left(z=L\right)$$

In Equation (19), χ_y0_ is the definition of the amount removed at z = L

The system of pressure and flow rate equations was solved analytically. Thus, only one ODE required a numerical solution. The gradient and Jacobian of this ODE were calculated explicitly. The initial value problem for each guess taken by the shooting method was started at z = L, meaning the dialysate inlet instead of the blood inlet. This reduced the magnitude of the initial values and ensured that the system’s variable (amount of toxin removed from blood) could not increase without bound. Large real eigenvalue components drive stiffness, so this minimizes it. This last modification, coupled with all the others, enabled stable solutions for the range of conditions tested.

A shooting method solves a boundary value problem (BVP) by solving a series of initial value problems (IVPs) that are numerically tractable. One variable is known at the starting boundary. For example, this could be dialysate toxin concentration at the dialysate inlet. The other variable (blood toxin concentration) is unknown. The variable χ (amount removed from blood) is zero at the blood inlet and unknown at the dialysate inlet. Guesses for the unknown value concentration are made within a reasonable parameter range until the known value (blood toxin concentration at the blood inlet) is obtained as the solution to the IVP.

Unmodified, a shooting method is inefficient because the plausible parameter space is very large, and small steps must be taken. Traditional methods of increasing efficiency, such as Regula Falsi or Newton’s method cannot cope with numerical instability due to stiffness. The analytical solution should only have one value of χ(z = L) that corresponds to χ = 0 at z = 0 (a zero-crossing). However, numerical instability causes multiple zero crossings, some of which are spurious. The modified shooting method uses predictable patterns in this system of equations to improve efficiency. Equally spaced guesses for the unknown variable (outlet blood toxin concentration) are made. The distance between the numerical solution and the known value (inlet blood concentration) is calculated and denoted Δ for each guess. The method searches for the value of the unknown variable at z = L that produces the known variable at z = 0. Thus, any pair of guesses that may contain a zero value of Δ in between them form an interval. Figure 2, panel A depicts a set of guesses with numerical instability. Values shown in red and blue are values of Δ. Figure 2, panel B illustrates nine types of intervals that may contain a zero value of Δ when some instability is present. Table 3 summarizes these interval types and how they are prioritized for subsequent searching. We design the algorithm to implement a breadth-first search of high-priority intervals to avoid long delays from spurious zero crossings.

Once intervals in a range of initial conditions are identified, they are queued. High-priority intervals are processed before low-priority intervals. Within each category, intervals are sorted by the distance from their bounds to the target (the known dialysate inlet toxin concentration). All initial conditions are tested, and the initial conditions whose bounds are closest to the target are selected. High-priority intervals are searched with a larger step size (since they contain no or limited instability), while low-priority intervals if searched, are searched with a smaller step size to ensure small “islands of numerical stability” near the true zero are not missed. A breadth-first search for the high-priority intervals is implemented: up to 13 intervals are searched sequentially before new intervals are queued. This minimizes computational resources wasted on searching spurious intervals. Figure 3 illustrates this process.

### 2.8. Numerical Implementation

The model was implemented using Julia Version 1.9.3 [40] with the packages Revise.jl, XLSX.jl, DifferentialEquations.jl [41], FileIO, JLD2, ODEInterfaceDiffEq [41], ODEInterface [41], NonlinearSolve [42], NLSolve [43], LinearAlgebra, and Roots [44]. We used the Julia function linspace provided by Jonathan Bieler [45]. We used the Julia remove function provided by Michael Franco [46]. We used the myfind Julia function [47]. The radau solver was used for the spatial solution [48]. The built-in solver selection algorithm of DifferentialEquations.jl was used for the temporal differential equation solution [49].

Parameter sweeps were conducted on the University of Washington’s Hyak Supercomputer. This computer uses the Rocky 8 operating system and the slurm scheduler. Individual nodes have 28 threads and 192 GB of memory.

The goodness of fit was analyzed by minimizing the sum of squares (goodness of fit to the entire solution) or by minimizing the percent error at the end of the trial (goodness of fit to the equilibrium bilirubin concentration). The two criteria are shown in Equations (20) and (21), respectively. Here, *t* is time during the trial, $t_{end}$ is the time at the end of the trial, $C_{true}$ is the measured concentration, $C_{model}$ is the concentration predicted by the model, and *i* is an index that varies from 1 to *n* where *n* is the number of time points.
(20)$$Error={\sum_{i=1}^{n}\left(C_{model}\left(t=t_{i}\right)-C_{true}\left(t=t_{i}\right)\right)}^{2}$$

Equation (20) (sum of squares goodness of fit criterion).
(21)$$Error=\frac{\left(C_{model}\left(t=t_{end}\right)-C_{true}\left(t=t_{end}\right)\right)\times 100\%}{C_{true}\left(t=t_{end}\right)}$$

Equation (21) (percent error goodness of fit criterion).

### 2.9. Step Size Sensitivity Test (Methods)

To verify that the model used a sufficiently fine mesh, tests were performed in which the maximum step size was decreased. The maximum step size along the z axis had been set to 1 mm for all modeling. A step size of 0.1 mm was tested, and the results were compared. In time, a maximum step size of 100 s was set for all modeling. A maximum step size of 10 s was tested for comparison.

### 2.10. Experimental Procedure

Five sets of experiments were carried out in total:-Condition 1: A pilot study (*n* = 3) using bovine serum albumin (BSA) on both sides of an F6HPS dialyzer (Fresenius, Waltham, MA, USA) with blood and dialysate flow rates of *Q_b_* = 180 mL/min and *Q_d_* = 90 mL/min. This was used to fit two parameters:•1: The dialyzer diffusive mass transfer coefficient for free bilirubin moving through polysulfone (*K_free_A_Q_d_=500_*)•2: The bilirubin binding equilibrium constant for the primary binding site on BSA (*kB,BSA*).-Condition 2: A validation data set (*n* = 3) using BSA on both sides of an F3 dialyzer (Fresenius, Waltham, MA, USA) with blood and dialysate flow rates of 150 mL/min.-Condition 3: A test data set (*n* = 3), like condition 2, but with the dialysate flow rate set to 20 mL/min, as previously described in a pilot study of a novel albumin dialysis system [23].-Condition 4: A test data set (*n* = 3), like condition 2, but with the dialysate flow rate set at 800 mL/min, which is the highest flow rate identified in clinical practice [50].-Condition 5: A test data set (*n* = 3), like condition 2, but with the dialysate flow rate set at 2 mL/min, which is relevant in some neonatal dialysis applications [51].•The parameter *β_Q_d__*, which measures the dependence of transmembrane toxin transport on flow rate, was fit once all five conditions were analyzed.

The precise values for the setups are summarized in Table 4. Concentrations are given as mass divided by volume in deciliters, abbreviated as dL. The F6HPS trials (setup 1) were used to set model parameters. These parameters were the toxin binding affinity for albumin, and the dialyzer mass transfer-area coefficient for free bilirubin diffusion at 500 mL/min dialysate flow rate ($K_{free}A_{Qd=500}$). The F3 trials were used to independently validate model predictions. The $K_{free}A$ value was adjusted between the two dialyzers (see Section 2.6). Flow rate was set using Masterflex Pumps with model numbers 07551-20 and 77521-40, using Easy-Load II Rotors with Model Number 772990-62 (Cole-Parmer, East Bunker, CT, USA) except for the 2 mL/min trial, in which an Ismatec 78017-10 (IDEX Corporation, Northbrook, IL, USA) pump was used on the dialysate side. Pressure was controlled using four RSCDRRE015PGSE3 pressure sensors (Honeywell International Inc., Charlotte, NC, USA) to prevent ultrafiltration, except in the 800 mL/min trial, where more robust PS04-G100KP-A4W pressure sensors (Same Sky, Lake Oswego, OR, USA) were used. Data from Honeywell pressure sensors were obtained using a Raspberry Pi 3B (Raspberry Pi, Cambridge, UK). The same Sky pressure sensor readings were processed using an HX711 (Sparkfun, Niwot, CO, USA) connected to an Arduino Uno (Arduino SA, Chiasso, Switzerland). The pressure was adjusted using c-clamps. The blood and dialysate reservoirs were immersed in a 37 °C water bath (Benchmark, Sayreville, NJ, USA), except in Setup 1. There, temperature was measured by K-type thermocouples connected to a Measurement Advantage DAQ and maintained between 35 and 40 °C on both sides using hotplates. Hotplates were from Corning (Glendale, AZ, USA), with model numbers PC-420, and PC-420D. K-type thermocouples were from MN Measurement Instruments (St. Paul, MN, USA). The DAQ was from Measurement Computing (Norton, MA, USA). Figure 4 shows diagrams of the circuits for each setup.

Solute concentrations are listed in Appendix A. pH on the blood and dialysate side was maintained between 7.35 and 7.45 using a pH Meter (OrionStar A211, Thermo Scientific, Waltham, MA, USA), except in Setup 1, where the pH was maintained between 7.2 and 7.5 using the less precise pHoenix XL meter from MesaLabs. For Setup 1, pH in this was initially set at 7.2 ± 0.1 following previous work on protein-bound toxin removal [17]. Acidosis is common in CKD patients [52,53]. However, pH values below 7.3 are unusual. Acknowledging this, later trials used a pH set point for Setup 1 of 7.4 ± 0.1. In all setups, pH adjustment was made using 1 N HCl and NaOH. pH was set at 35–40 °C, determined using hotplates and thermocouples as previously described. In all setups, care was taken to protect the setup from light to minimize photodegradation of bilirubin. For Setup 1, dialyzers were reused after being cleaned according to established clinical protocols [54,55]. Bilirubin and Albumin measurement protocols were the same as in previous work [23].

**Table 4 bioengineering-11-01262-t004:** Albumin dialysis setups used in this study.

	Setup 1	Setups 2–5
Duration (h)	3	5
Blood analog solution volume (mL)	628.33	200
Dialysate volume (mL)	626.67	200
Blood flow rate (mL/min)	180	150
Dialysate flow rate (mL/min)	90	2, 20, 150, 800
Ultrafiltration rate (mL/min)	0	0
Blood bovine serum albumin (BSA) concentration (g/dL)	2	2
Dialysate BSA concentration (g/dL)	2	2
Dialyzer	F6HPS	F3
Material	Polysulfone	Polysulfone
Number of fibers	8400 [56]	2304 [56]
Hollow fiber inner radius (µm)	100 [57]	100 [58]
Hollow fiber outer radius (µm)	140 [57]	140 [59]
Hollow fiber length (cm)	21 [60]	20 [58]
Area (m^2^)	1.3 [57]	0.4 [58]
Housing inner radius (mm)	20 [57]	11 (measured)
Bilirubin reflection coefficient (assumed [17])	0	0

### 2.11. Polysulfone Membrane Hydraulic Permeability

Polysulfone membrane hydraulic permeability was measured by a method based on previous studies [61], with some modifications. A small dialyzer, called a “mini-module” was created out of individual F6HPS dialyzer fibers in a custom-made casing. Figure 5 shows the test setup. The formula for hydraulic permeability was:(22)$$L_{p}=\frac{∆V/∆t}{P\times A}$$

∆V refers to the ultrafiltration volume. ∆t is the time. *A* is the total area of the dialysis membrane in the mini-module. Here, pressure is calculated as:(23)$$P=\frac{P_{b,in}+P_{b,out}}{2}-\frac{P_{d,in}+P_{d,out}}{2}$$

In this formula, $P_{b,in}$ is the inlet blood pressure and $P_{b,out}$ is the outlet blood pressure. $P_{d,in}$ and $P_{d,out}$ are the corresponding dialysate pressures. Since the module is near the surface of a stirred water bath and the dialysate side is in contact with the water (Figure 5), the dialysate pressures are approximately equal and negligible.

Mini-modules were constructed out of F6HPS dialyzer polysulfone fibers. Fibers were carefully selected to be straight, to avoid any damage or kinking. Two mini-modules were constructed (designated F6HPS and F6HPS2). They were then bundled together in a fitting using Gorilla Epoxy (Gorilla Glue, Cincinnati, OH, USA). Each mini-module used 10 fibers. Its total area was 9.42 cm^2^. Fibers were 0.15 m long, with a 100 µm inner radius and a 140 µm outer radius.

A neMESYS low pressure syringe pump (Cetoni, Korbussen, Germany) was used to set precise flow rates. Pressure on the inlet and outlet of the mini-module was measured using PX409-USBH sensors (Omega Engineering, Norwalk, CT, USA). Mini-modules were immersed in a 37 °C water bath. Heating, stirring, and temperature control were provided using hotplates, K-type thermocouples, and a data acquisition unit (DAQ) as previously described. Flow through the mini-module was compared to flow through a control tube. The difference was the ultrafiltration rate ∆V. Following past work [61], flow rates of 1910 $\frac{\mu L}{\min}$, 764 $\frac{\mu L}{\min}$, and 191 $\frac{\mu L}{\min}$ were tested for both mini-modules. One of the 191 $\frac{\mu L}{\min}$ tests was an extreme outlier, so it was repeated, and 382 $\frac{\mu L}{\min}$ was also tested for that mini-module. These additional tests did not replicate the outlier value, so the outlier value was then discarded.

## 3. Results

### 3.1. Non-Replication of the Model of Pei and Colleagues

Applying the parameters they provided, the work of Pei and colleagues [14,15,16,17] could not be replicated. They reported results for three initial blood bilirubin concentrations: 14.7, 17.7, and 21.4 mg/dL. Their reported modeled final blood bilirubin concentrations are 7.7, 9.0, and 10.8 mg/dL. Attempting to reproduce their model with their parameters yielded final bilirubin concentrations of 10.36, 11.08, and 11.76 mg/dL. It is possible that a difference in the definition of the Peclet number accounts for this discrepancy since Pei and colleagues do not provide this parameter in their manuscripts. The inability to replicate the previous work of Pei and colleagues prompted the development of the model presented in this work.

### 3.2. Hydraulic Permeability

Hydraulic permeability was measured as 8.61 × 10^−11^ m/(s×Pa) ± 2.14 × 10^−11^ m/(s×Pa) (average ± standard deviation). This measurement was used for subsequent modeling. This is much lower than the value obtained by Liao and colleagues for polysulfone of 1.628 × 10^−9^ m/(s×Pa) [61]. However, it is comparable to the value obtained by Benavente and Jonsson of 10^−10^ m/(s×Pa) [62]. The raw data for the hydraulic permeability test is reported in Appendix A. The discarded outlier test is highlighted in bold.

### 3.3. Albumin Dialysis Study

Table 5 summarizes the measured average starting bilirubin and albumin concentrations for all five setups. These values were used for modeling to avoid errors caused by variations in the initial solution composition.

### 3.4. Step Size Sensitivity Test (Results)

The impact of reducing the maximum spatial step was minimal. We compared results for all five conditions when the step size was set to 1 mm or 0.1 mm or when the maximum time step was decreased from 100 s to 10 s. The largest difference was a 0.032% change for condition 4, when the spatial step size was decreased. This demonstrates that the mesh applied was sufficiently fine to prevent errors due to discretization.

### 3.5. Kf_ree_A_Qd=500_ and kB Parameter Sweep

The optimal values of *kB,BSA* and $K_{free}A_{Qd=500}$ for bilirubin removal Setup 1 were found to be *kB,BSA* = 0.5E7 (1/M) and $K_{free}A_{Qd=500}$ = 2500 mL/min. This result was not sensitive to changing the number of fibers from *n* = 8400 provided by Fresenius to *n* = 9200 [57]. The impact of this change was −0.018%. The results are shown in Figure 6. Panel A shows the parameter sweep results for the sum of squares criterion. Panel B shows the parameter sweep results for the percent error criterion. Figure 7 shows the model prediction compared to the true measurements.

### 3.6. Model Validation for F3 Dialyzer and Flow Rates

The model was validated using Condition 2, then tested against the other three flow rates for the F3 polysulfone dialyzer. Applying Equation (20), the new $K_{free}A_{Qd=500}$ for this dialyzer was 769.23 mL/min. Figure 8 panel A shows the model prediction for different values of $\beta_{Q_{d}}$. All values of the parameter fit the data for 150 mL/min and 800 mL/min dialysate side flow rates. This validates the $K_{B}$ and $K_{free}A_{Qd=500}$ values obtained from the F6HPS dialyzer. However, for lower flow rates, models with $\beta_{Qd}$ values of 0.3 and less predict a less than 5% change in bilirubin removal across the entire range of flow rates tested. This deviates from our observation. Applying the sum of squares goodness of fit criterion, $\beta_{Qd}=0.45$ fits the data best. This is shown in Figure 8, panel B. This means a 45% decline in $K_{free}A_{Qd=500}$ when the dialysate side flow rate declines by 300 mL. Individual conditions are shown in Figure 9, along with model predictions for $\beta_{Qd}=0$ (no dependency on flow rate), $\beta_{Qd}=0.05544$ (literature value for urea), and $\beta_{Qd}=0.45$ (best fit for bilirubin data).

Table 6 summarizes the accuracy of the model in predicting the outcomes in these conditions. For all flow rates measured, the model predicted that the final blood bilirubin concentration would be within 6%. Further study of whether $\beta_{Qd}$ varies between dialyzers is warranted since applying the $\beta_{Qd}=0.45$ model to the original F6HPS condition increased the error.

## 4. Discussion

We have developed a thermodynamically-based computational model that can be used to rationally design albumin dialysis. This paves the way for optimization of an albumin dialysis system for protein-bound toxin removal. It also creates the potential for personalized dialysis regiments, where patients are given flow rate conditions and dialysate compositions tailored to their body volume and toxin concentrations. This would play a similar role to the NxStage dosing calculator in home dialysis [63]. This work will guide us in optimizing AMOR treatment prescriptions.

This model differs from previous models since it accounts for both ultrafiltration and backfiltration, in addition to diffusive transport and protein binding. It considers differences in toxin transport throughout the length of the dialyzer rather than treating it as a well-mixed compartment or averaging ultrafiltration across the entire membrane. It does not make any assumptions about the albumin and toxin concentration. It also incorporates the dependence of membrane permeability on flow rate. A novel algorithm was used to improve its numerical stability over a wide range of test conditions.

This work analyzed the model’s ability to predict the impact of changes in dialyzer and dialysate flow rate. We demonstrate that there is a pronounced effect of dialysate flow rate on bilirubin removal at low flow rates (2 mL/min–20 mL/min), which becomes insignificant at greater flow rates (150 mL/min and 800 mL/min). When the flow rate changes by 300 mL/min, the membrane transfer coefficient for free bilirubin changes by 45%. We estimate this effect using a linear approximation based on prior work on urea [26]. This effect may be due to the formation of a boundary layer on the dialysate side. Further work using computational models that explicitly model the boundary layer, such as that of Snisarenko and colleagues [64], would test this proposed mechanism.

With high flow rate conditions (150 mL/min and 800 mL/min), our model has been validated against conditions that were not used to set its parameters. This is strong evidence of its predictive power. The flow rate adjustment is necessary for lower flow rates (2 mL/min and 20 mL/min). Additional independent validation is needed to confirm the value of the flow rate adjustment parameter. In their original work on the subject, Depner and colleagues noted a small but statistically significant variability of $\beta_{Qd}$ between dialyzers [26]. Other dialyzers or other flow rates may provide independent verification of our $\beta_{Qd}$ value. When we applied the best-fit model for the F3 dialyzer to the F6HPS data, error increased. This may indicate this parameter varies between dialyzers.

A limitation of this work is that it is “post-blind” [65]. Our pre-blind analysis is available in the form of a preprint [66]. A numerical error in that version of the model caused us to underestimate the deviation at 20 mL/min. Additional external validation in which the version of the model presented here is the “pre-blind” version will increase confidence in this modeling approach.

Further work is in progress to predict the impact of changes in blood flow rate and other parameters. Another limitation was the small scale. In the test conditions used to validate the model, blood volume was 200 mL–630 mL. In a patient, the plasma volume is approximately 3 L [67]. A larger scale in vitro test is needed to validate the model’s predictive ability in patients.

The model correctly predicts the equilibrium outcomes of albumin dialysis but overestimates the kinetics of bilirubin transport across the membrane at high dialysate flow rates. This appears mathematically in an implausibly high $K_{free}A_{Qd=500}$ value. For example, it predicts a $K_{free}A_{Qd=500}$ value for bilirubin of 2500 mL/min for the F6HPS dialyzer, whose *kA* for urea is only 746 mL/min under similar flow conditions [68]. Pei and colleagues previously reported a bilirubin *kA* value of 800 mL/min for a Gambro 6LR dialyzer using a similar model [17]. The *KoA* for urea for the Gambro 6LR dialyzer is 736 mL/min [38]. We could not replicate their result with an 800 mL/min *kA* using their model and Villarroel’s definition of Peclet Number [20]. A higher *kA* value would be needed. In their original work on protein-bound toxin removal modeling, Patzer and Bane noted that the dialyzer mass transfer coefficient of their membranes increased after 180 min (3 h) of dialysis [5]. They suggested that bilirubin and albumin binding to membrane pores accounts for this phenomenon. It would be very useful to accurately predict bilirubin kinetics in our model so that it can be coupled with models of bilirubin absorption onto activated charcoal to predict the behavior of a combined dialysis/absorption system like MARS. Thus, we will extend the model to incorporate bilirubin binding to the dialysis membrane. Snisarenko and colleagues have presented a theoretical framework for this in their proposed dialysis membrane design [64].

Another potential extension of the model is to incorporate non-Newtonian blood rheology [69]. For this work, a blood analog solution consisting of albumin and toxins dissolved in dialysate was used. However, if plasma or whole blood were used instead, the impact of non-Newtonian rheology would need to be considered. The model’s equations can be modified to incorporate a different definition of viscosity.

## 5. Conclusions

We present a computational model that uses the fundamental physics of flow rate, pressure, and concentration in the dialysis process to predict the rate of protein-bound toxin removal using albumin dialysate. We validate this model against experimental data at four different flow rates and for two different polysulfone dialyzers. For all conditions, final bilirubin concentrations are predicted to have an error of less than 6%. This demonstrates the value of this model for designing optimal albumin dialysis protocols. We use this modeling to determine the impact of the dialysate side flow rate on the transport of free bilirubin across the dialyzer membrane. The following linear model best fits our data: when the dialysate flow rate declines by 300 mL/min, the membrane transfer coefficient $K_{free}A$ declines by 45%. Experiments with additional dialyzers and flow rates should be undertaken to validate this conclusion.

## 6. Patents

This work is covered by the provisional patent “Computational Model of Sorbent Dialysis—[IP: 50020.01US1]”.

## Figures and Tables

**Figure 1 bioengineering-11-01262-f001:**
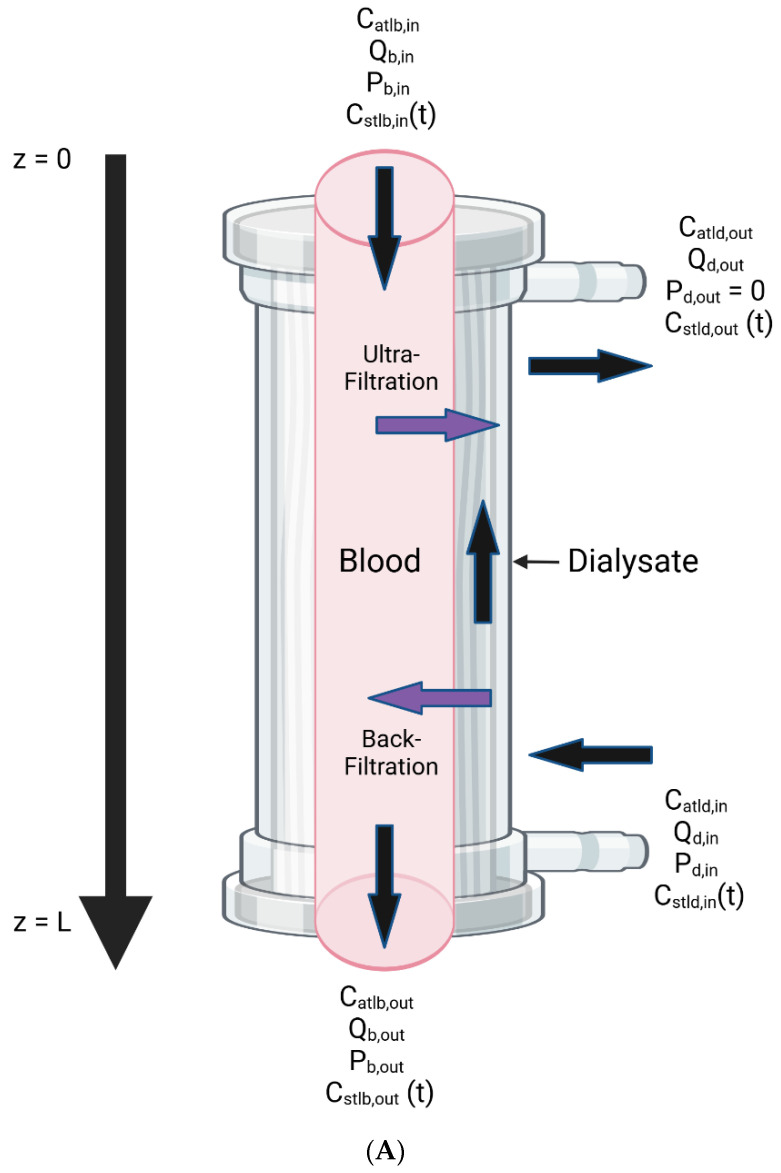
(**A**) Model flow directions, boundary conditions, and variables (Figure created with BioRender). (**B**) Local ultrafiltration and backfiltration influenced by pressure under conditions of net zero ultrafiltration. Arrows show the direction of fluid movement across the dialyzer membrane, with upward arrows indicating ultrafiltration and downward arrows indicating backfiltration. (Figure created with BioRender).

**Figure 2 bioengineering-11-01262-f002:**
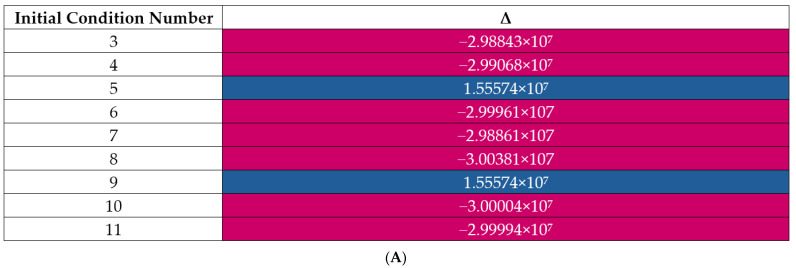
(**A**) Values of the distance between the initial value problem solution and the known blood inlet concentration value at z = 0, denoted Δ for a set of z = L initial conditions. An interval is a pair of values that may contain a value whose solution has Δ = 0 between them. (**B**) How each interval type may contain the true zero. Red dots represent sampled conditions (since not all values of the dialysate outlet concentration are sampled). Blue rectangles represent unstable areas. The green rectangle represents the correct target subinterval. Figure created with BioRender.

**Figure 3 bioengineering-11-01262-f003:**
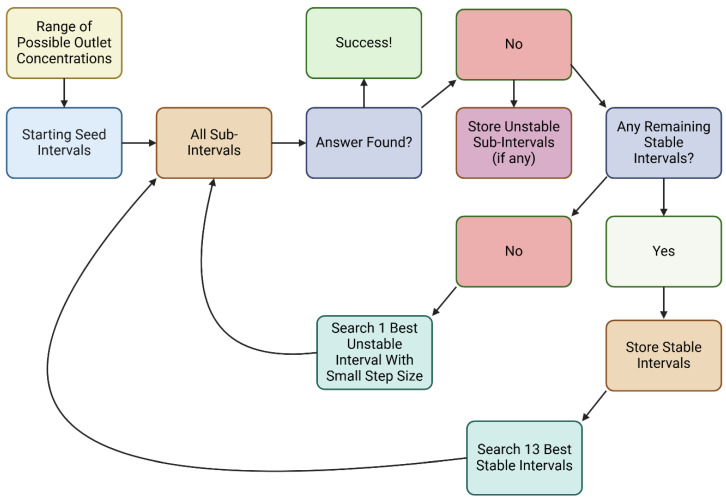
Flow Chart of the interval search process. Created in BioRender.

**Figure 4 bioengineering-11-01262-f004:**
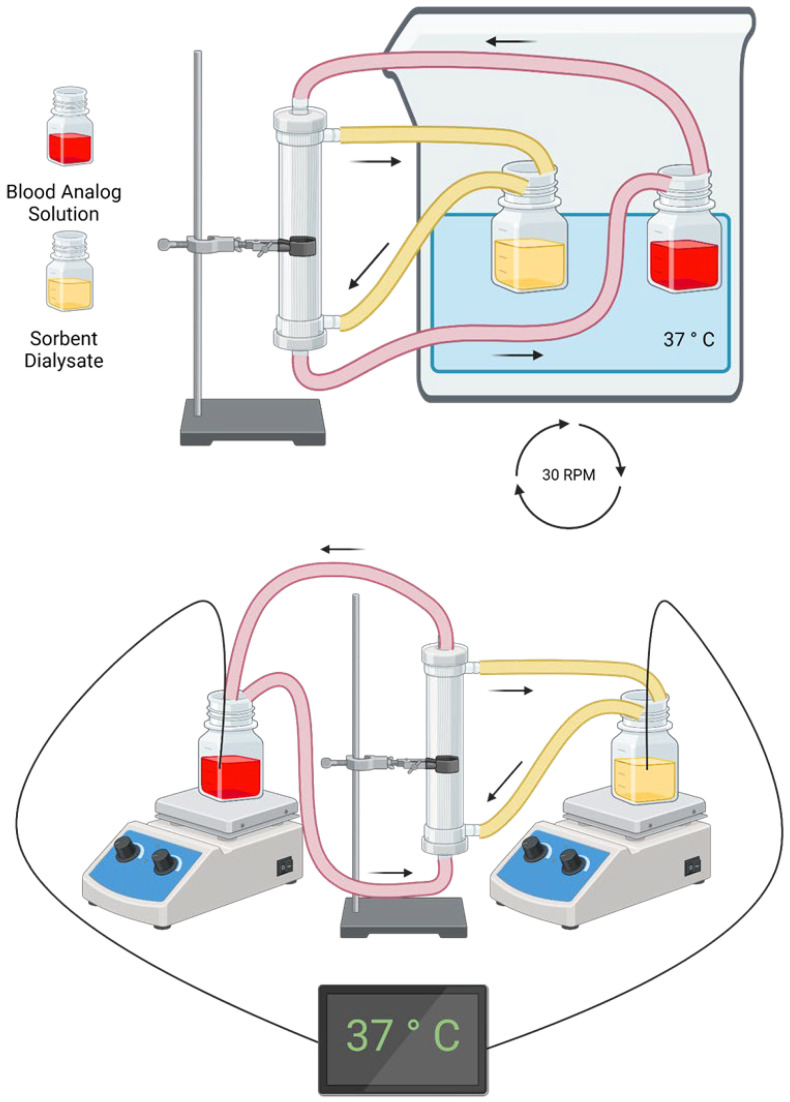
Experimental setup for conditions 2–5 (**top**) and condition 1 (**bottom**). Figure created with BioRender.

**Figure 5 bioengineering-11-01262-f005:**
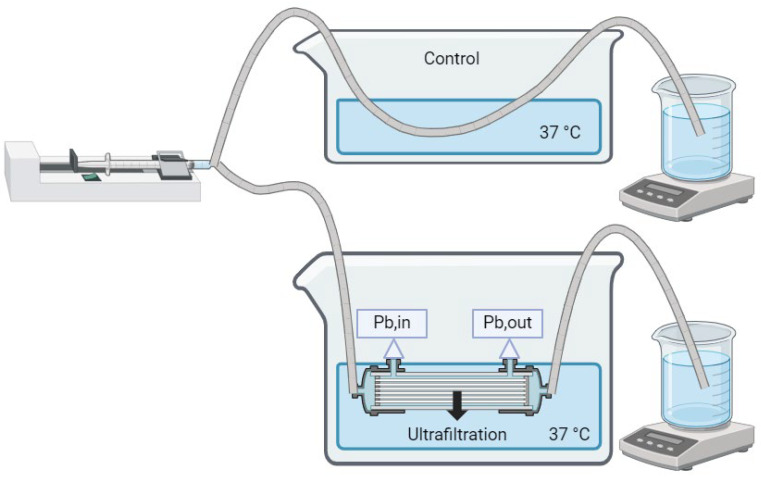
Hydraulic permeability test setup. Figure created with BioRender.

**Figure 6 bioengineering-11-01262-f006:**
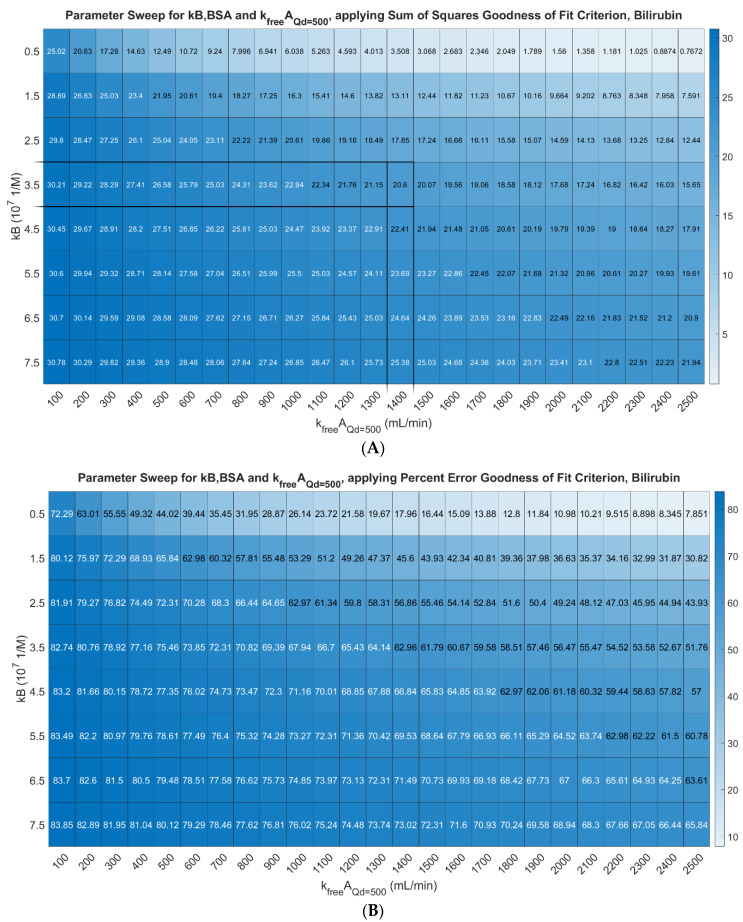
Parameter sweep results for *kB,BSA* for bilirubin, and $K_{free}A$, measuring best fit by two criteria. (**A**): Sum of squares criterion. (**B**): Percent error criterion.

**Figure 7 bioengineering-11-01262-f007:**
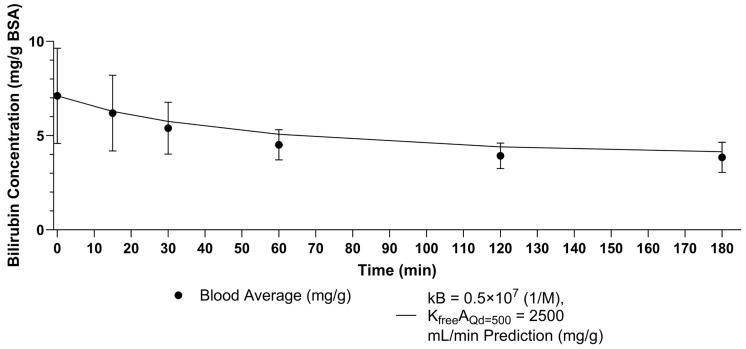
Blood side reservoir bilirubin over time for F6HPS with *kB* = 0.5×10^7^ (1/M) and $K_{free}A_{Qd=500}$ = 2500 mL/min and *n* = 8400 compared to experimental data. Error bars are standard deviation. Final percent error 7.85%. Sum of squares error 0.77.

**Figure 8 bioengineering-11-01262-f008:**
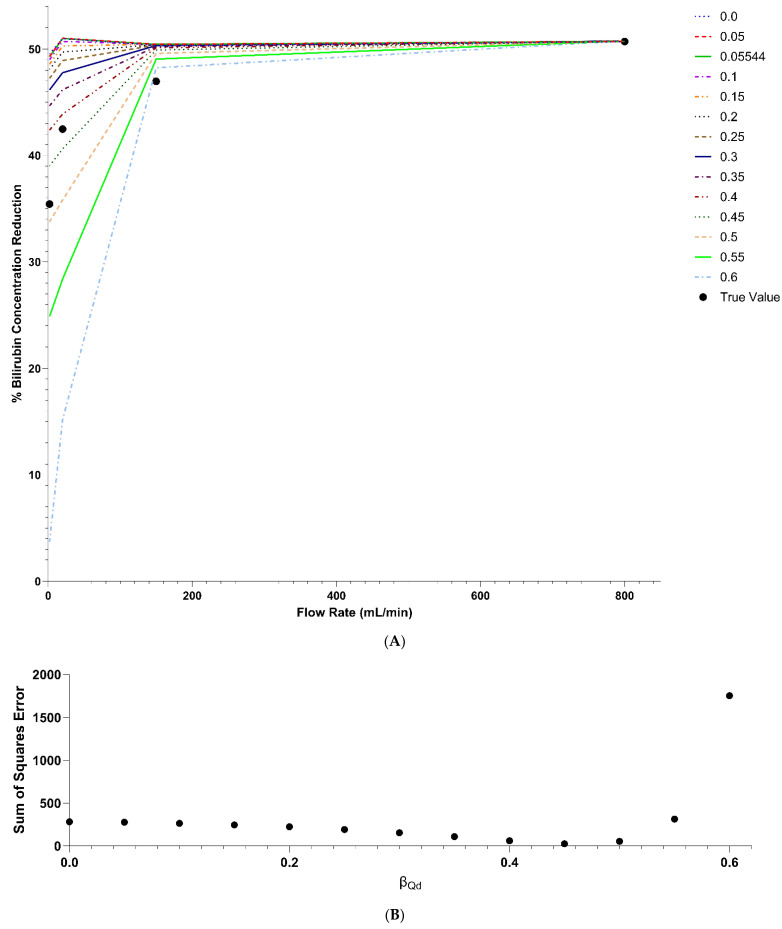
Model predictions for bilirubin removal with varying $K_{free}A$ dialysate side flow rate dependency parameters (**Panel A**). The sum of squares error for fitting the observed relationship between percentage concentration decline and dialysate side flow rate (**Panel B**).

**Figure 9 bioengineering-11-01262-f009:**
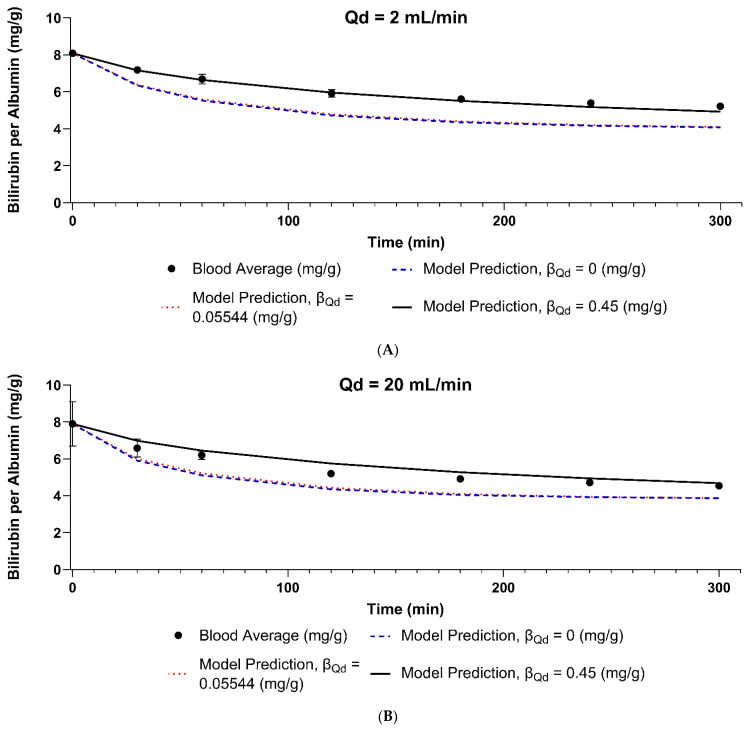
Results and model predictions for dialysate flow rates of 2 mL/min (**Panel A**), 20 mL/min (**Panel B**), 150 mL/min (**Panel C**), and 800 mL/min (**Panel D**). Errors bars are standard deviation. Where error bars are not shown, it is because they would be smaller than the data point depicted on the graph. Points are experimental data, and the lines are the model predictions.

**Table 1 bioengineering-11-01262-t001:** Summary of existing models of closed-loop mode albumin dialysis. Only assumptions regarding the albumin circuit are listed. Assumptions regarding patients (for clinical systems) and absorbent columns (for MARS) are not listed.

Study	Transport Phenomena	Equations	Assumptions Regarding Albumin Circuit	Boundary and Initial Condtions	Validation and Fitting
Magosso et al. [8]	1: Ultra-filtration 2: Diffusion 3: Association and dissociation	1: Two-compartment model of patient 2: Equation with diffusive and convective transmembrane transport and sorbent.	1: Well-mixed compartments, including dialyzer 2: One toxin–albumin binding site 3: Constant ultrafiltration 4: Osmosis is negligible	1: Measured concentrations at t = 0 2: Bound and unbound toxin at equilibrium at t = 0	1: Two parameters fit on one session and extended to all sessions 2: Three parameters set for each treatment 3: Eight sessions modeled
Annesini et al. [9,10,11,12,13]	1: Diffusion 2: Association and dissociation	1: One-compartment model of patient 2: Albumin circuit described following Patzer and colleagues [4,5,6] 3: Linear driving force mass transfer kinetics for MARS sorbent columns	1: Well-mixed compartments, except dialyzer 2: One toxin–albumin binding site 3: Albumin concentration >> toxin concentration 4: Osmosis is negligible	1: Measured concentrations at t = 0 2: Bound and unbound toxin at equilibrium at t = 0 3: Dankwerts conditions for sorbent column model	Patients and in vitro data. Sample size not given.
Pei et al. [14,15,16,17]	1: Ultra-filtration 2: Diffusion 3: Association and dissociation	1: Initial value problem describes toxin removal over time. 2: Boundary value problem describes toxin removal over a single pass through the dialyzer	1: Well-mixed compartments, except dialyzer 2: One toxin–albumin binding site 3: Osmosis is negligible 4: Zero net ultra-filtration	1: Measured concentrations at t = 0 2: Bound and unbound toxin at equilibrium at t = 0 3: Inlet and outlet concentration. Regula Falsi for boundary value problem.	Three in vitro tests were used to fit parameters

**Table 2 bioengineering-11-01262-t002:** Reported values of bilirubin–BSA binding constant.

Study	Primary Bilirubin–BSA Binding Constant (1/M)	Temperature (°C)	Solution	pH
Chen, 1973 [29]	2.2 × 10^7^	25	0.1 M potassium phosphate buffer	7.4
Faerch and Jacobsen, 1974 [30]	2.7 × 10^7^	37	67 mM phosphate buffer	7.4
Rubaltelli and Jori, 1979 [31]	2.16 × 10^7^	19	0.5 M KH_2_PO_4_-Na_2_HPO_4_ buffer	7.4
Muzaffar et al., 1991 [32]	1.05 × 10^7^	30	Tris-HCl buffer	8.0
Williams et al., 2002 [33]	1.2 × 10^7^	Room temperature	0.125 M phosphate buffer	7.4
Chen et al., 2007 by fluorescence enhancement [34]	6.57 × 10^7^	Not given	Double-distilled water	Not given
Chen et al., 2007 by fluorescence quenching [34]	4.34 × 10^7^	Not given	Double-distilled water	Not given

**Table 3 bioengineering-11-01262-t003:** Interval types for modified shooting method.

Interval Type	Action
Type 1: Negative to positive transition	Consider as high-priority for further searching
Type 2: Positive to negative transition	Same as Type 1
Type 3: Negative to positive with unstable values in between	Consider as Low Priority. Search only after high-priority intervals have been searched. Use a smaller step with more subintervals to search for stable subintervals. Once a subinterval is found, proceed as if Type 1.
Type 4: Positive to negative with unstable values in between	Same as Type 3
Type 5: Negative to unstable, with no positives after it	Same as Type 3
Type 6: Unstable with no negatives before it to positive	Same as Type 3
Type 7: Positive to smaller positive	Same as Type 1 (without instability, dialysate inlet concentration would increase whenever dialysate outlet concentration increases, with all other parameters held constant. Thus, this indicates instability and potentially zero crossings in the subintervals)
Type 8a: Positive to smaller positive with instability in between that does not encompass the entire interval	Same as Type 1
Type 8b: Positive to smaller positive with instability in between that encompasses the entire interval	Same as Type 3 (to avoid an infinite loop)

**Table 5 bioengineering-11-01262-t005:** Average initial bilirubin and albumin concentration. *n* = 3 for all conditions. Values are shown as mean ± standard deviation.

Condition	Starting Blood Bilirubin (mg/dL)	Starting Blood Albumin (g/dL)	Starting Dialysate Albumin (g/dL)
1 (F6HPS, *Qb* = 180 mL/min, *Qd* = 90 mL/min)	15.37 ± 3.06	2.24 ± 0.37	1.77 ± 0.41
2 (F3, *Qb* = *Qd* = 150 mL/min)	18.81 ± 2.84	1.90 ± 0.07	1.94 ± 0.23
3 (F3, *Qb* = 150 mL/min, *Qd* = 20 mL/min)	16.76 ± 2.35	2.13 ± 0.12	2.29 ± 0.13
4 (F3, *Qb* = 150 mL/min, *Qd* = 800 mL/min)	18.28 ± 0.43	2.03 ± 0.02	2.10 ± 0.04
5 (F3, *Qb* = 150 mL/min, *Qd* = 2 mL/min)	17.36 ± 1.94	2.15 ± 0.22	2.21 ± 0.14

**Table 6 bioengineering-11-01262-t006:** Goodness of fit for different conditions by sum of squares and percent error of final value Criteria.

Condition	$\mathrm{Sum}\;\mathrm{of}\;\mathrm{Squares},\;\beta_{Qd}=0$	$\mathrm{Percent}\;\mathrm{Error}\;\mathrm{of}\;\mathrm{Final}\;\mathrm{Value},\;\beta_{Qd}=0$	Sum of Squares, $\beta_{Qd}=0.05544$	Percent Error of Final Value, $\beta_{Qd}=0.05544$	Sum of Squares, $\beta_{Qd}=0.45$	Percent Error of Final Value, $\beta_{Qd}=0.45$
1 (F6HPS, *Qb* = 180 mL/min, *Qd* = 90 mL/min, training)	0.566	6.99	0.767	7.85	5.70	25.1
2 (F3, *Qb* = *Qd* = 150 mL/min, validation)	5.69	−6.35	5.31	−6.58	1.69	−5.55
3 (F3, *Qb* = 150 mL/min, *Qd* = 20 mL/min, test)	4.21	−14.8	3.61	−14.8	0.719	3.23
4 (F3, *Qb* = 150 mL/min, *Qd* = 800 mL/min, test)	4.12	−2.79	4.48	−2.82	6.67	−2.94
5 (F3, *Qb* = 150 mL/min, *Qd* = 2 mL/min, test)	7.86	−21.9	7.24	−21.4	0.147	−5.58

## Data Availability

The data and code presented in this study are available on request from the corresponding author due to our software being patent-pending.

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
