# Peer review of "Predicting the Impact of Polysulfone Dialyzers and Binder Dialysate Flow Rate on Bilirubin Removal"

_bioengineering, 2024, doi:10.3390/bioengineering11121262_

Round 1
Reviewer 1 Report
Comments and Suggestions for Authors
This paper concerns the simulation of the effect of dialysate flow rate on bilirubin removal efficiency in binder dialysis. I believe that this study contributes to the development of binder dialysis. However, there are a few points of concern, and I offer my comments below. I hope my comments will help improve the manuscript.
1. Title: The title includes the words "dialyzer choice," but since only polysulfone membranes are used, I think " dialyzer choice" is inappropriate.
2. Related to 1., it would have been nice to see research on membrane materials other than polysulfone.
3. Equation 4: as in reference [16], but this equation represents the opposite flow from equation 3, so I think that (Pb-Pd) is correct for (Pd-Pb).
4. Please specify the size and the number of hollow fibers of the "mini-module", as well as other specifications.
5. Table 3: The starting dialysate albumin in Condition 5 is written differently from the others.
Author Response
Dear Editors and reviewers: The attached document "Revision Letter.docx" contains our replies to all reviewer comments and revisions. Because significant revisions were made and there was overlap between reviewer comments, we have chosen to present our replies in the form of a single document for convenience.
The replies to reviewer 1 have been highlighted

Reviewer 2 Report
Comments and Suggestions for Authors
The bioengineering-3307980 manuscript presents a computational model that uses the fundamental physics of flow rate, pressure, and concentration in the dialysis process to predict the rate of protein bound toxin removal using albumin dialysate. The manuscript is consistent with the journal's focus, but it contains a number of shortcomings.
Please avoid personal pronouns.
Line 61 "Unfortunately, when we replicated their model, we were unable to reproduce their results." This is incorrect. This phrase is only appropriate if you have performed a simulation experiment and shown differences in the results in a published paper.
The manuscript contains a number of common phrases whose meaning is unclear:
It is not entirely clear what "It accounts for ultrafiltration and diffusion" means in terms of modeling. This occurs several times. It is clear how to account for diffusion transfer. What does it mean to account for ultrafiltration? Do you mean convective transfer? Please clarify. In the introduction, please write specifically for other models what types of transfer were taken into account, what equations, assumptions, and boundary conditions were used.
Similar comments to the phrase "This model applies chemical engineering techniques." What does this mean in terms of modeling?
The novelty of the work is not clear. If new mathematical approaches are proposed, they are not reported. The problem statement is not compared with the described works of other researchers.
The physical meaning of some parameters is unclear: coefficient representing the diffusive transport of unbound bilirubin across the entire area of the dialyzer membrane; Albumin-bilirubin binding constant; local ultrafiltration flux. How are they determined? Fitting parameters should be indicated at the first listing of model parameters. Item 2.6 should be placed immediately after the listing of parameters.
2.6. Fit Parameters. The authors indicate the range for the fitting parameters and justify the choice of their numerical values. However, it is not entirely clear why they refuse the literary values used by other researchers.
The range from 100 to 2500 mL/min for another parameter is still large. Be more specific. Why do you start from 100 mL/min?
Sorry, I don't understand why concentrations are expressed in mg/dL. What does dL mean?
A diagram needs to be added to illustrate the boundary conditions.
More details are needed for the phrase "Parameter sweeps were conducted on the University of Washington's Hyak Supercomputer." Specify the computing power and other parameters.
The quality of the figures needs to be improved. Figure 6 is clearly of poor quality. The scales are small, the symbols are difficult to read.
The conclusions should contain specific results, preferably those that are quantifiable.
Comments on the Quality of English Languageno specific comments
Author Response
Dear Editors and reviewers: The attached document "Revision Letter.docx" contains our replies to all reviewer comments and revisions. Because significant revisions were made and there was overlap between reviewer comments, we have chosen to present our replies in the form of a single document for convenience.
The replies to reviewer 2 have been highlighted

Reviewer 3 Report
Comments and Suggestions for Authors
1. In my opinion, a diagram with flow notations and boundary conditions is missing to better describe the mathematical model. I recommend that the authors create such a diagram.
2. The mathematical model needs a more detailed description. In this form, it is described very poorly. It is said that the removal of the toxin through the dialyzer was solved at each time step by solving a spatial model of concentrations, pressures, and flow rates. At the same time, the equations of a one-dimensional model are given below. It is not clear what spatial model we are talking about? How is this spatial model related to the one-dimensional one?
3. Blood tends to exhibit non-Newtonian properties, especially in very narrow membrane channels. How does the model take rheology into account? This is very important.
4. Flow in very narrow fiber channels is considered. Can we assume that blood flow is described by the continuous medium equations used in the work?
5. The numerical solution of the system of differential equations presented in the work tends to exhibit numerical instability. How do the authors overcome this problem. It is necessary to provide data from methodical calculations proving the stability of the numerical scheme. It is also unclear whether the authors have studied the effect of the size of the time and coordinate splitting step. This is very important.
6. The authors provide all the results in dimensional form. At the same time, there is no attempt to generalize the results. I propose that when considering the dependence on consumption, use the values reduced to consumption (blood bilirubin concentration) / (dialysate side flow rate), etc.
Author Response
Dear Editors and reviewers: The attached document "Revision Letter.docx" contains our replies to all reviewer comments and revisions. Because significant revisions were made and there was overlap between reviewer comments, we have chosen to present our replies in the form of a single document for convenience.
The replies to Reviewer 3 have been highlighted.

Round 2
Reviewer 2 Report
Comments and Suggestions for Authors
Dear Authors,
thank you for your answers. However, they are extremely confusing. Please prepare answers for each reviewer separately. Due to changes in the MDPI system, we do not see other reviewers' comments. Only our own. I have written to the MDPI editors about this more than once, but I have not received an answer. Having only answers to other reviewers' comments without their comments creates chaos that is difficult to understand. At the same time, this also makes it easy to hide some of the comments and not respond to them.
Even if you have removed some information in the manuscript that was the subject of a reviewer's comment, please take the time to note this.
In addition, you still have not answered my question regarding the terminology that remains in the manuscript. It is not entirely clear what "It accounts for ultrafiltration and diffusion" means in terms of modeling. This occurs several times. What does it mean to account for ultrafiltration? Do you mean convective transfer? Please clarify. You wrote that you made a replacement and now call it ultrafiltration rate. Although in fact you use different terms, including local ultrafiltration and simply "ultrafiltration". At the same time, in the Model Description you use the term "convective flux". What does this mean? How is the ultrafiltration rate different from taking into account the convective flux? Aren't they the same thing?
What does the coefficient reflecting "The percent change in 𝐴 from 500 to 800 mL/min" mean? How is it selected and verified?
"This parameter was initially set to 0.05544 based on past work on urea" References are needed! Does this make physical sense?
Please add more clarity and specificity to the description of Figure 2. Now it looks like an explanation that can be left in the text, but not in the title.
"Intervals were divide into nine types and prioritized". Please add more clarity. What intervals are we talking about? The meaning is lost. It needs to be explained
The table numbering is wrong.
What is the point of writing that the model takes into account the Ultrafiltration rate (mL/min), if Table 1 (according to the authors’ numbering) indicates that it was set equal to 0?
Comments on the Quality of English Languageno specific comments
Author Response
Dear Reviewer #2,
We thank you for the second round of reviewer comments. Our replies are as follows:
With regards to the first comment on accounting for ultrafiltration:
- We are referring to local ultrafiltration. Ultrafiltration rate is in mL/min. Convective flux is Jv, which has units of mm2/s. We realize that there is a typo in equation 6: The units of the correction factor should be 1/mm instead of mm3/s. We have corrected this on page 6, line 158 of the revised manuscript.
Regarding the second question: “What does the coefficient reflecting "The percent change in Kfree? from 500 to 800 mL/min" mean? How is it selected and verified?” and “"This parameter was initially set to 0.05544 based on past work on urea" References are needed! Does this make physical sense?”
- We explain this on page 7, line 166. The reference is reference 26, which appears in the previous paragraph. To improve clarity, we added references to reference 26 on line 166, page 7 and line 214 on page 8. To clarify the physical meaning of this coefficient we added the phrase “This indicates a 5.544% change in mass transfer area coefficient during a 300 mL/min change in dialysate side flow rate.” On page 7, line 166.
Regarding the third question: “Please add more clarity and specificity to the description of Figure 2. Now it looks like an explanation that can be left in the text, but not in the title” and the question “"Intervals were divide into nine types and prioritized". Please add more clarity. What intervals are we talking about? The meaning is lost. It needs to be explained”
- An additional panel was added to Figure 2 showing an example of an interval with numerical instability. This is on page 10, line 278. The description in the text, on page 9, line 254, was updated.
Regarding the fourth question: “The table numbering is wrong.”
- We corrected table and figure numbering.
Regarding the fifth question: “What is the point of writing that the model takes into account the Ultrafiltration rate (mL/min), if Table 1 (according to the authors’ numbering) indicates that it was set equal to 0?”
- We clarify that we are referring to local ultrafiltration and backfiltration. A second panel is added to Figure 1 demonstrating local ultrafiltration and backfiltration. This is on page 5, line 140. The new panel is introduced on page 3, lines 137-138.
In addition to edits that addressed reviewer feedback we corrected a typo in the title of section 3.5, page 18, line 434 and a typo in the description of Figure 8 (previously misnumbered Figure 5) on page 470, line 475.
We welcome any further feedback you may have.
Thank you again,
Sincerely,
Alexander Novokhodko
Reviewer 3 Report
Comments and Suggestions for Authors
The authors have substantially revised the paper and responded to all my comments. The article may be published.
Author Response
Dear Reviewer,
Thank you very much for your attention to this matter. I appreciate your feedback.
Sincerely,
Alexander Novokhodko
Round 3
Reviewer 2 Report
Comments and Suggestions for Authors
Dear Authors, thank you for your patience and more detailed information. Although I am not completely satisfied with the visual content of the manuscript (different styles of tables, graphs and figures from one to another), I must admit that the manuscript has become more understandable in content.